# Development of Imeglimin Electrospun Nanofibers as a Potential Buccal Antidiabetic Therapeutic Approach

**DOI:** 10.3390/pharmaceutics15041208

**Published:** 2023-04-11

**Authors:** Ali A. Alamer, Nasser B. Alsaleh, Alhassan H. Aodah, Abdullah A. Alshehri, Fahad A. Almughem, Sarah H. Alqahtani, Haya A. Alfassam, Essam A. Tawfik

**Affiliations:** 1Advanced Diagnostics and Therapeutics Institute, Health Sector, King Abdulaziz City for Science and Technology (KACST), Riyadh 11442, Saudi Arabia; 2Department of Pharmacology and Toxicology, College of Pharmacy, King Saud University, Riyadh 12372, Saudi Arabia

**Keywords:** buccal delivery, fast dissolving fibers, electrospinning, imeglimin, diabetes

## Abstract

The prevalence of type 2 diabetes (T2D) has been growing worldwide; hence, safe and effective antidiabetics are critically warranted. Recently, imeglimin, a novel tetrahydrotriazene compound, has been approved for use in T2D patients in Japan. It has shown promising glucose-lowering properties by improving pancreatic beta-cell function and peripheral insulin sensitivity. Nevertheless, it has several drawbacks, including suboptimal oral absorption and gastrointestinal (GI) discomfort. Therefore, this study aimed to fabricate a novel formulation of imeglimin loaded into electrospun nanofibers to be delivered through the buccal cavity to overcome the current GI-related adverse events and to provide a convenient route of administration. The fabricated nanofibers were characterized for diameter, drug-loading (DL), disintegration, and drug release profiles. The data demonstrated that the imeglimin nanofibers had a diameter of 361 ± 54 nm and DL of 23.5 ± 0.2 μg/mg of fibers. The X-ray diffraction (XRD) data confirmed the solid dispersion of imeglimin, favoring drug solubility, and release with improved bioavailability. The rate of drug-loaded nanofibers disintegration was recorded at 2 ± 1 s, indicating the rapid disintegration ability of this dosage form and its suitability for buccal delivery, with a complete drug release after 30 min. The findings of this study suggest that the developed imeglimin nanofibers have the potential to be given via the buccal route, thereby achieving optimal therapeutic outcomes and improving patient compliance.

## 1. Introduction

Diabetes remains among the leading causes of morbidity and mortality, resulting in enormous burdens on healthcare systems worldwide [1]. The global prevalence of type 2 diabetes (T2D) has increased over the past few years [1]. A recent study has found that such an increase in the prevalence of T2D is mainly due to urbanization and improved socioeconomic status [2]. Although public health measures, such as education, public awareness, and implementation of healthy lifestyle changes, are critical for mitigating this global healthcare challenge, there is yet a need for the development of novel therapies for the treatment of T2D with better control of blood glucose levels and minimal adverse responses [3]. Furthermore, emerging perspectives are focused on tackling the challenge of obesity to treat T2D. Hence, unravelling novel therapeutics that could regulate blood glucose levels and improve cellular energetics is of utmost importance [4].

Several promising therapeutics for the treatment of T2D have been developed over the past recent years, and among these is imeglimin [3]. It is a tetrahydrotriazene-containing small molecule that belongs to a novel class of oral antidiabetics known as glimins [5]. In June 2021, imeglimin was approved for use in T2D patients in Japan, and several emerging research studies provided evidence for its efficacy as monotherapy or in combination with insulin in T2D patients [6,7,8]. Imeglimin mediates its glucose-lowering effect through multiple mechanisms, including enhancing pancreatic beta-cell function and preserving its integrity, suppressing liver gluconeogenesis, and improving insulin sensitivity in the liver and skeletal muscles [9]. The underlying molecular mechanism of imeglimin action is mainly mediated by improving mitochondrial function and suppressing mitochondrial-derived free radicals [9,10]. Interestingly, recent research demonstrates the protective properties of imeglimin beyond its glucose-lowering effect across various cell types and tissues, as demonstrated in several experimental disease models, such as cardiovascular and neurodegenerative [11,12,13]. Indeed, such findings may extend the future use of imeglimin beyond its current indication in T2D patients.

Treatment with imeglimin has demonstrated good safety and efficacy outcomes [6,7,14,15]. Nevertheless, several drawbacks have been reported, including reduced absorption with increased doses, potentially due to saturation of active transport and gastrointestinal (GI) discomfort, such as nausea and vomiting, constipation, and diarrhea [14,15,16]. On the other hand, the population of geriatric patients (aged 60 years and over) is growing worldwide, and it is estimated to be 1.4 and 2.1 billion by 2030 and 2050, respectively [17]. Multiple chronic diseases commonly occur in geriatric patients requiring various medications, which may lead to poor medication compliance [18]. In addition, geriatric patients may experience age-related issues concerning achieving optimal therapeutic outcomes, including difficulty swallowing and reduced GI function [19,20].

Similarly, the pediatric population may experience similar problems when achieving optimal therapeutic outcomes [19]. Such issues could be overcome by developing novel formulations via different routes of administration and utilization of advanced techniques in pharmaceutical technology, such as smart buccal delivery systems [19]. Furthermore, the emerging applications of imeglimin beyond its glucose-lowering properties, for example, the improvement of mitochondrial function and its consequent protection against different diseases, further emphasize the importance of developing novel formulations.

Oral and buccal delivery has several advantages, including ease of administration, fast absorption, bypassing first-pass metabolism, and low GI absorption [21,22]. Different fast-dissolving mucoadhesive pharmaceutical formulations have been established, such as transdermal patches, tablets, gels, and ointments, with the thin oral and buccal patches offering ease of administration and fast rate of absorption (i.e., seconds) [21,22]. Nevertheless, these thin oral films have limitations, including dose uniformity and film thickness, which could alter mechanical properties and drug release rates [23]. Smart buccal delivery systems utilize novel and advanced materials to achieve optimal absorption and drug delivery to its ultimate target [24,25]. Indeed, smart buccal delivery systems may overcome several limitations of thin oral and buccal film formulations [24,25]. Electrospun nanofibers are a key smart buccal delivery system with several advantages, including unique mechanical properties and ultra-rapid disintegration and drug release rates (≤2 s) [26]. Electrospinning is a widely used method to make nanofibers by applying high voltage to a capillary tube of polymer solution to form electrostatic repulsion between surface charges, eventually forming nanofibers [27]. Electrospun nanofibers have a wide range of applications, including biomedical applications such as drug delivery, wound dressing, and tissue engineering [27]. Electrospun nanofibers have high porosity and surface-to-volume ratio, allowing a large volume of drug loading and fast release and absorption of the loaded drug [26,28].

The current study is aimed to fabricate electrospun imeglimin nanofibers with the ultimate goal of having a novel imeglimin formulation that provides rapid disintegration and absorption rates via the buccal route. The imeglimin nanofibers will be prepared by the electrospinning technique and characterized for their fibrous diameter, drug-loading, disintegration and drug-release profile. The fabricated imeglimin-loaded nanofibers will have several advantages over conventional imeglimin, including ease of administration and patient compliance in geriatric and pediatric populations and avoidance of GI-related adverse responses. In addition, the nanofiber formulation will have other benefits, such as faster absorption rates, the onset of action, and bypassing drug first-pass metabolism in the liver.

## 2. Materials and Methods

### 2.1. Materials

Polyvinylpyrrolidone (PVP) with an average molecular weight of ~1,300,000, ethanol (≥99.5%, HPLC grade), methanol (HPLC grade), acetonitrile (HPLC grade), and phosphate buffer saline tablets (PBS; pH 7) were supplemented from Sigma-Aldrich (St. Louis, MO, USA), while triethanolamine was purchased from Loba Chemie (Mumbai, India). PBS solution was prepared by adding 5 tablets in one-liter distilled water and mixing until completely dissolved; then, the pH was adjusted to 6.8 using 5M hydrochloric acid. Imeglimin hydrochloride (CAS No. 775351-61-6) was bought from Med Chem Express (Monmouth Junction, NJ, USA). Imeglimin (purity: 99.98%) has a molecular weight of 191.66 and a water and DMSO solubility of ≥50 mg/mL and 25 mg/mL, respectively [29]. The chemical structure of this drug is shown in the Appendix A. Distilled water was generated through Milli Q, Millipore (Billerica, MA, USA).

### 2.2. Preparation of Imeglimin-Loaded Nanofibers

Imeglimin-loaded nanofibers fabrication was performed using the Spraybase^®^ electrospinning setup (Dublin, Ireland), according to a modified method [28]. To prepare the PVP polymer solution, PVP powder was initially dissolved in absolute ethanol with 8% (*w*/*v*) concentration and stirred at room temperature for 2 h until the complete dissolving of the PVP polymer. The imeglimin was mixed with the polymer solution in a 1% (*w*/*v*) concentration and stirred for an hour to allow for homogeneity between the drug and the polymer solution. PVP solution, alone as 8% (*w*/*v*), without adding the drug, was used as a blank (i.e., drug-free) formulation. The electrospun nanofibers were formulated at room temperature and 30–45% relative humidity. The electrospinning setup syringe pump was adjusted at a flow rate of 800 µL/h, a needle-to-collector distance of 15 cm, and an inner needle diameter of 0.9 mm. To achieve a stable jet, a voltage in the 7–9 kV range was applied, and the surface of the metallic collector was covered with aluminum foil to collect the prepared nanofibers. PVP is a water-soluble, non-toxic, biocompatible polymer approved for use in the pharmaceutical industry by the United States Food and Drug Administration (US FDA). Hence, it is widely used in medicine and cosmetics [30].

### 2.3. Morphology and Diameter Assessment of the Electrospun Nanofibers

The prepared blank and drug-loaded nanofibers were characterized, in terms of surface morphology and diameter, using the scanning electron microscopy (SEM) instrument (JSM-IT500HR SEM, JEOL Inc., Peabody, MA, USA) at a 5 kV accelerating voltage and average measurements of 70 fibers for each fibrous system. The fabricated nanofibers were collected directly onto aluminum foil, and platinum (2 nm) was used to coat the samples in a JEC-3000FC auto fine coater (JEOL Inc., Peabody, MA, USA). The nanofibers diameter analysis was measured using ImageJ software (National Institute of Health, Bethesda, MD, USA).

### 2.4. X-ray Diffraction (XRD) Analysis

The XRD analysis of imeglimin, PVP, the physical mixture (PM; equivalent amount to the drug-loaded fibers for the drug and polymer mix as raw materials), blank, and drug-loaded fibers was performed using Rigaku Miniflex 300/600 (Tokyo, Japan). Cu Kα radiation (λ = 1.5148 227 Å) was equipped in the XRD instrument, with a current and voltage of 15 mA and 40 kV, respectively. All tested formulations were placed on glass holders and scanned between 2θ of 5° to 50° at a scan speed of 5°/min. OriginPro^®^ 2020 software (OriginLab Corporation, Northampton, MA, USA) was employed to plot the patterns of XRD results.

### 2.5. Disintegration Test of the Electrospun Nanofibers

The blank and drug-loaded nanofibers disintegration was evaluated following a modified petri dish method [23]. A total of 3 mg of the nanofibers squares were placed into a petri dish containing pre-warmed (37 °C) PBS, with a pH adjusted to 6.8 to mimic the oral cavity pH. This was performed in a thermostatic shaking incubator (Excella E24 Incubator Shaker Series, New Brunswick Scientific Co., Enfield, CT, USA). Time recording of the complete fibrous mat detachment was measured, and the data are presented as the mean ± standard deviation (SD) of three independent replicates.

### 2.6. Determination and Quantification of Imeglimin

Imeglimin was quantified using the Waters e2695 HPLC system that consists of a Waters^®^ 717 plus autosampler, Waters 600 binary pump, and Waters 2489 UV/detector (Waters Technologies Corporation, Milford, MA, USA). Imeglimin was separated by an isocratic elution of a mobile phase of triethanolamine (1%, adjusted by formic acid to pH 3.9), methanol, and acetonitrile at a ratio of 60%, 5%, and 35%, respectively, and a Waters XSelect HSS Cyano C_18_ column (4.6 mm × 250 mm, 5 μm), with a temperature that was maintained at 20 °C. The flow rate of the mobile phase was adjusted at 1 mL/min, the injection volume was kept at 10 μL, and the detection was achieved at a wavelength of 254 nm.

A stock solution of imeglimin was prepared by dissolving the drug into PBS (pH 6.8). A serial dilution ranging from 100 to 1.56 μg/mL was used to develop the imeglimin calibration curve according to the method mentioned above. The drug retention time (Rt) was achieved at 3.3 min.

### 2.7. Determination of Drug Loading (DL) and Fiber’s Yield (Y) of Imeglimin-Loaded Nanofibers

The DL of imeglimin-loaded nanofibers was measured by dissolving at least three pieces of the nanofibers (4.1 ± 0.1 mg) in 5 mL PBS (pH 6.8), and each solution was maintained at room temperature for 7 h to ensure complete nanofibers dissolution. The concentration and amount of imeglimin were measured using the developed HPLC method of detection, and the following equation calculated the DL:(1)$$DL=\frac{Entrapped\;drug\;amount}{Yield\;of\;nanofibers\;amount}$$

The following equation was applied to measure the Y of the imeglimin-loaded fibers:(2)$$Y=\frac{Actual\;nanofibers\;amount\;}{Theoretical\;nanofibers\;amount\;}\times 100$$

The number of theoretical nanofibers was quantified based on the number of solid materials (drug and polymer) in the total volume of the spinning solution. All the results are presented as mean ± SD of three independent replicates.

### 2.8. Determination of the Drug Release of Imeglimin-Loaded Nanofibers

The release profile of imeglimin-loaded nanofibers was identified by dissolving at least three pieces of the fibrous mat (9 ± 1 mg) into 10 mL of pre-warmed PBS (pH 6.8) in glass vials. A thermostatic shaking incubator (Excella E24 Incubator Shaker Series, New Brunswick Scientific Co., Enfield, CT, USA) was used to determine the drug release of imeglimin-loaded nanofibers, with a temperature and shaking rate of 37 °C and 100 RPM, respectively. A total of 1 mL of the tested formulation was withdrawn at a predetermined time-point, ranging from 1 to 180 min, and substituted with an equivalent volume of pre-warmed PBS (pH 6.8). The imeglimin amount was quantified using the developed method of HPLC, and the cumulative release percentage was calculated as a function of time by the following equation:(3)$$Cumulative\;Release\;\%=\frac{Cumulative\;drug\;amount}{Theoretical\;drug\;amount\;}\times 100\;$$

The results are presented as the mean ± SD of at least three independent replicates.

The release of imeglimin-loaded nanofibers was also assessed by Franz diffusion (PermeGear, Hellertown, PA, USA). A piece of a semipermeable membrane, with a cut-off from 12 to 14 KD (spectra/por^®^, San Francisco, CA, USA), was placed in between a donor and receptor chamber of the Franz diffusion apparatus. The latter chamber held 3 mL of pre-warmed PBS (pH 6.8), and the experiment was performed at 37 ± 0.5 °C with continuous stirring. At least three pieces of the fibrous mat (8 ± 0.5 mg) were placed into the donor chamber, and 0.5 mL PBS (pH 6.8) was added to dissolve the nanofiber mat. A total of 100 μL samples were withdrawn from the receptor compartment at a predetermined time-point ranging from 15 min to 24 h and substituted with an equivalent volume of pre-warmed PBS (pH 6.8). The imeglimin amount was quantified using the developed method of HPLC, and the cumulative release percentage was calculated as a function of time by Equation (3).

### 2.9. In Vitro Cytotoxicity Assessment of Imeglimin

The in vitro cytotoxicity of imeglimin was assessed using the MTS assay according to the modified method [31]. The MTS reagent (cell Titer 96^®^ Aqueous one solution cell proliferation assay) was purchased by Promega (Southampton, UK). It measured the cellular metabolic activity of human fibroblast HFF-1 cells (ATCC- SCRC-1401) following a 24 h exposure to the tested drug. HFF-1 cells were exploited in this study since it was previously reported as an in vitro model for drug delivery research of the oral mucosal cavity [32,33,34]. Cells were sub-cultured in Dulbecco’s modified eagle medium (DMEM), contained 10% (*v*/*v*) fetal bovine serum (FBS) and additional essential components for media preparation; all were obtained from Sigma-Aldrich (St. Louis, MO, USA).

Trypan blue exclusion test was used for cell counting, and the cells were seeded at a seeding density of 1.5 × 10^4^ cells per well into a 96-well microtiter plate. HFF-1 cells were kept in the incubator at 37 °C and 5% CO_2_ overnight. A total of 100 µL of increasing dose of the imeglimin, from 15.6 to 1000 µg/mL, were exposed to the HFF-1 cells for 24 h. Cells incubated with DMEM only or 0.2% Triton X-100 were used as positive and negative controls, respectively. The consumed medium was aspirated from each well, and 100 µL of fresh DMEM plus 20 µL of the MTS reagent were added into the wells. The cells were incubated for 3 h at 37 °C and 5% CO_2_. A Cytation 3 microplate reader (BIOTEK instruments inc, Winooski, VT, USA) was used at 492 nm to measure the absorbance of MTS, and the below equation calculated the percentage of cell viability:(4)$$Cell\;Viability\;\%=\frac{\left(S-T\right)}{\left(H-T\right)\;}\times 100$$
where *S* is the absorbance of HFF-1 cells treated with imeglimin, *T* is the absorbance of HFF-1 cells treated with Triton X-100, and *H* is the absorbance of HFF-1 cells treated with cell culture media. The results are presented as the mean ± SD of at least three replicates.

### 2.10. Statistical Analysis

OriginPro^®^ 2020 software (OriginLab Corporation, Northampton, MA, USA) was used to analyze the data statistically and plot the graphs presented in this study.

## 3. Results and Discussion

### 3.1. Morphology and Diameter Assessment of the Electrospun Nanofibers

The morphology analysis of the blank and imeglimin-loaded nanofibers using SEM exhibited that the fabricated nanofibers had smooth, non-porous, and non-beaded surfaces, as shown in Figure 1a,b, respectively. The diameter measurement indicated average fiber diameters of 450 ± 82 nm and 361 ± 54 nm for the blank and drug-loaded nanofibers, respectively (Figure 1). This slight difference in the diameter might be attributed to the higher voltage stabilizing the spinning jet, allowing for smaller diameter fibers [35]. The surfaces of imeglimin-loaded nanofibers showed no observed drug crystals. All morphological features suggest successful preparatory criteria consistent with previous PVP nanofibers systems [26,28,36,37].

### 3.2. X-ray Diffraction (XRD) Analysis

The physical state of raw materials (PVP, imeglimin, and PM), blank, and imeglimin-loaded nanofibers was examined using XRD and confirmed the solid dispersion of the drug owing to electrospinning. Figure 2 shows the intense Bragg reflections pattern of imeglimin at 2θ: 14.71°, 16.82°, 24.36°, 24.87°, 25.42°, 31.64°, and 34.03°, which indicate for the crystallinity nature of the drug. There were some reflections that appeared in the PM at almost similar 2θ angles (15.56°, 16.83°, 23.6°, 24.55°, 25°, and 33.85°) but in much less intensity than the pure drug, indicating the presence of meglumine in the crystalline form within the polymer mixture. On the other hand, no intense characteristic peaks of the PVP polymer and blank nanofibers were observed, which is expected due to the amorphous nature of the PVP polymer, as reported in several previous studies (Figure 2) [38,39]. The imeglimin-loaded nanofibers showed a broad halo, which confirms the amorphous transformation of the drug due to solid dispersion. It was previously reported that the electrospinning technique allows for solid dispersion of the loaded drugs [35]. This might be because of the solvent’s rapid evaporation, which may hinder the molecules’ organization into the crystalline lattice, as explained in [40].

### 3.3. Disintegration Test of the Electrospun Nanofibers

The disintegration test is considered an essential parameter for any oral cavity dosage forms such as sublingual and buccal medications, which should be ≤30 s, according to US FDA regulation [41]. Figure 3 demonstrated that the disintegration of the blank and imeglimin-loaded nanofibers was rapid in PBS (pH 6.8) and initiated from the edges to the center, causing shrinking of the fibrous mat until complete dissolving. The blank and imeglimin-loaded nanofibers were disintegrated at 1 ± 1 s and 2 ± 1 s, respectively, indicating the suitability of the fabricated nanofibers for buccal delivery. This ultra-rapid disintegration feature was also seen in previous studies utilizing PVP nanofibers [26,37].

### 3.4. Determination and Quantification of Imeglimin

Imeglimin quantification was performed by a developed HPLC, where it showed a successful drug separation after 3.3 min (i.e., Rt), as presented in the Appendix A. The regression equation and coefficient of determination (R^2^) were determined as y = 81462x + 119483 and 0.9992, respectively, as shown in Figure 4. The calibration curve indicated this developed method’s excellent linearity and successful drug separation.

### 3.5. Determination of Drug Loading (DL) and Fiber’s Yield (Y) of Imeglimin-Loaded Nanofibers

The DL of the imeglimin-loaded nanofibers was measured by the above-developed HPLC method, which was calculated as 23.5 ± 0.2 µg/mg of the fibers. The data indicated that the Y of the drug-loaded nanofibers was quantified as 78%, and most likely the remaining percentage was lost during the peeling of the fibrous mat off the aluminum foil and the deposition of some fibers onto the walls of the electrospinning instrument chamber. This fibrous loss was also seen in previous PVP electrospun fibers studies [42].

### 3.6. Determination of the Drug Release of Imeglimin-Loaded Nanofibers

The release of imeglimin was evaluated by placing the drug-loaded nanofibers in glass vials using custom-made sinkers to be sunk in the bottom of the vials owing to the floating nature of the fibrous mats. The imeglimin-loaded nanofibers exhibited a burst release in the first minute (i.e., 42%), followed by a cumulative drug release of 87% after 5 min, and a complete release of imeglimin appeared after 30 min, as presented in Figure 5a. It was expected that an ultra-rapid release profile due to the hydrophilicity of the PVP polymer, the molecular dispersion of imeglimin upon electrospinning the high surface-to-volume ratio of nanofibers, and the ultra disintegration of the nanofibers (i.e., 2 s), which enhanced the dissolution and accelerated the release profile of the loaded drug. Tort et al., Aburayan et al., and Tuğcu-Demiröz et al. demonstrated the rapid release profile of PVP nanofibers, who studied the release of ornidazole-, halicin-, and metronidazole-loaded PVP nanofibers, respectively [37,42,43].

Imeglimin release was also assessed by the Franz diffusion cell method. As shown in Figure 5b, the release of imeglimin was sustained from approximately 17% after 15 min to 69% after 6 h, followed by a complete drug release after 24 h. This sustained release pattern was expected due to the permeability of the drug within the semipermeable membrane of the Franz diffusion apparatus, which reduced the drug’s release rate considerably compared to the thermo-shaking incubator method, which measured the drug directly after its release from the nanofibers. In addition, owing to the high mucoadhesive property of PVP polymer [28,44], imeglimin was probably retained on the membrane up to 24 h, which could be an additional advantage of this drug delivery system by enhancing the permeability of the loaded drug through high polymer adhesion on biological membranes, namely the oral mucosa. Further permeability testing is required to evaluate the permeability of imeglimin-loaded nanofibers on an ex vivo membrane that represents the oral cavity compared to the free drug.

### 3.7. In Vitro Cytotoxicity Assessment of Imeglimin

Evaluating the loaded drug’s cytotoxicity is essential to assess its safety on living cells and prove its suitability for therapeutic applications. It is worth noting that owing to the FDA approval of PVP polymer for being considered a safe pharmaceutical excipient, this study did not assess the cytotoxicity of PVP. Different doses of imeglimin exposed to HFF-1 cells were evaluated over 24 h, and the result is demonstrated in Figure 6. Increasing the concentration of imeglimin showed no significant decrease in the cellular viability of HFF-1 cells. The concentrations ≥500 µg/mL exhibited a percentage of cellular viability below 80%, while the lowest applied dose (15.6 µg/mL) revealed a high rate of cellular viability (≈100%). This cytotoxicity profile of imeglimin indicates its safe application upon 24 h exposure on human skin fibroblast cells at all applied doses ≤250 µg/mL. Unfortunately, comparing the cytotoxicity data with the literature was difficult, as most work demonstrates the protective effects against glucose- or other toxicant-induced toxicity [11,45]. However, it is worth mentioning that exposure to high levels of imeglimin up to 100 µM (~20 µg/mL) for 24 h or 10 mM (~2 mg/mL) for 4 h was not associated with toxicity in human endothelial cells (HMEC-1) [11]. This result suggests that imeglimin can be considered safe in vitro. Further in vitro and in vivo studies are essential before claiming the suitability of this delivery system for buccal.

## 4. Conclusions

In sum, the carried-out assessments in this study have demonstrated that imeglimin was successfully loaded into ultra-rapid disintegrating PVP electrospun nanofibers. The formulated imeglimin nanofibers had smooth, non-porous, and non-beaded surfaces with an average diameter of 361 ± 54 nm. The nanofibers exhibited a DL of 23.5 ± 0.2 μg/mg of the fibers, rapid drug release of about 87% after 5 min, and a complete drug release after 30 min. Such quick release is attributed to the hydrophilic nature of PVP, the large surface-to-volume ratios of the nanofibers, and their ultra-rapid disintegration rates (≤2 ± 1 s). Together, this study provides a novel delivery platform of imeglimin nanofibers with the potential to be administered via the buccal route with the advantage of overcoming existing challenges of free drugs, including suboptimal oral absorption and GI-associated adverse responses. In addition, this formulation can potentially enhance drug compliance in special populations, including geriatrics and pediatrics. Future studies are warranted to evaluate and validate the formulated nanofibers in relevant in vivo models.

## Figures and Tables

**Figure 1 pharmaceutics-15-01208-f001:**
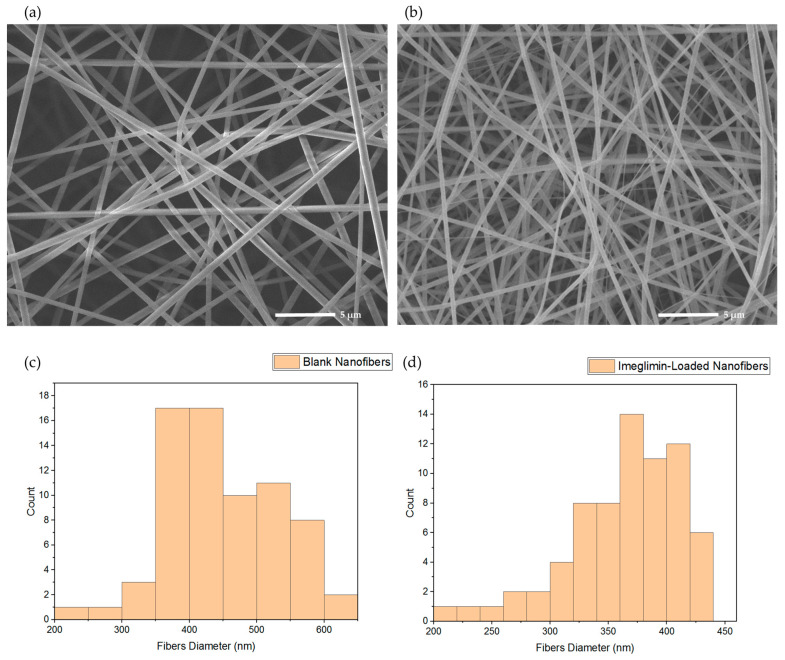
SEM images that show (**a**) blank nanofibers and (**b**) imeglimin-loaded nanofibers as smooth, non-porous, and non-beaded, with average fibers diameters of 450 ± 82 nm and 361 ± 54 nm, respectively (n = 70). The diameter distribution of (**c**) blank nanofibers and (**d**) imeglimin-loaded nanofibers show the average diameters of the fibrous systems.

**Figure 2 pharmaceutics-15-01208-f002:**
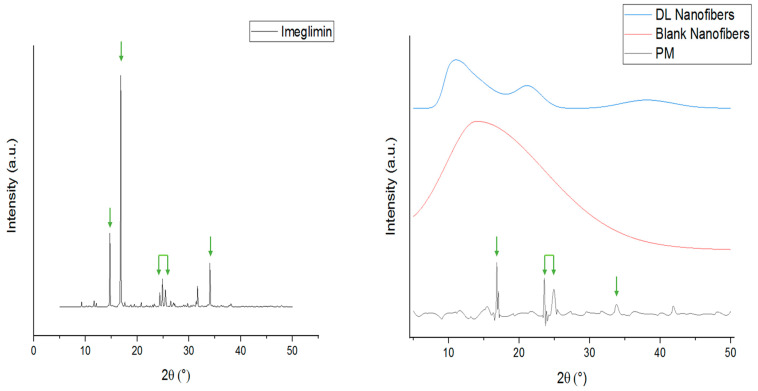
XRD patterns of imeglimin, PM, blank, and imeglimin-loaded nanofibers showing that imeglimin was in the crystalline form (presence of characteristic reflections at 2θ: 14.71°, 16.82°, 24.36°, 24.87°, 25.42°, 31.64°, and 34.03°, which are reflected by the green arrows) as pure and in the PM. The blank and drug-loaded nanofibers were amorphous (broad halo pattern). The absence of the imeglimin distinct peaks in the drug-loaded nanofibers suggests the molecular dispersion of the drug due to the electrospinning process. DL: drug-loaded, PM: physical mixture.

**Figure 3 pharmaceutics-15-01208-f003:**
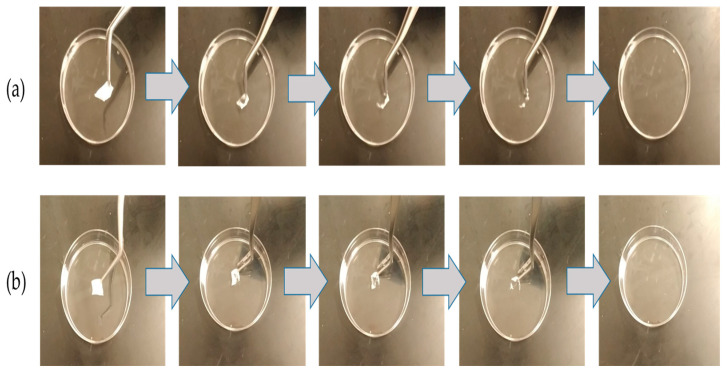
The disintegration of (**a**) blank nanofibers and (**b**) imeglimin-loaded nanofibers demonstrates that both nanofibrous systems dissolved ultra-rapidly (≤2 ± 1 s, n = 3).

**Figure 4 pharmaceutics-15-01208-f004:**
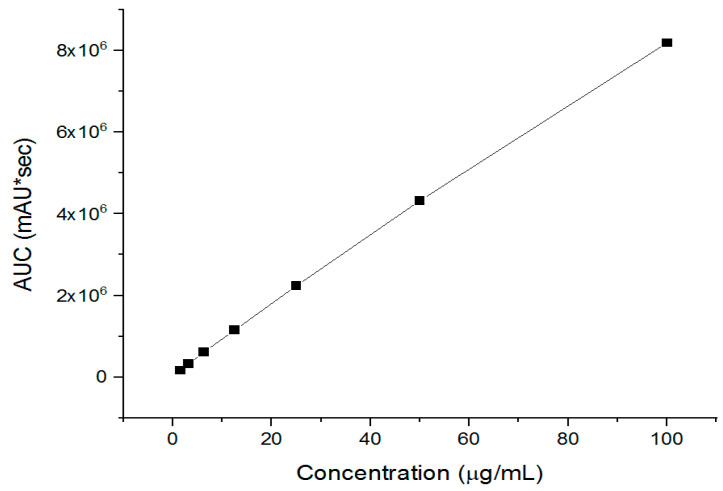
The developed HPLC calibration curves of imeglimin show its excellent linearity (R^2^ = 0.9992).

**Figure 5 pharmaceutics-15-01208-f005:**
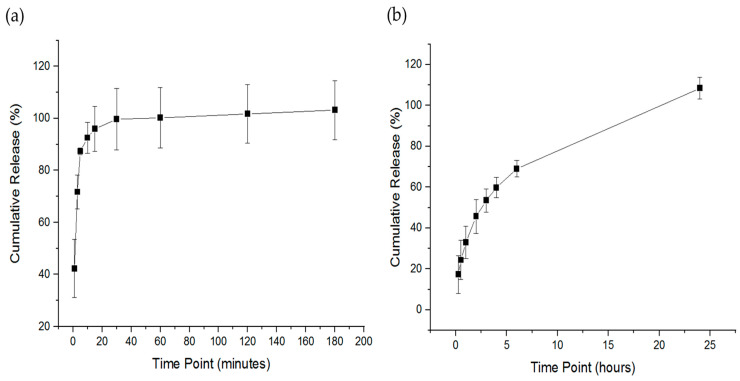
The release profile of the imeglimin-loaded nanofibers. The drug release profile of the thermo-shaking incubator method (**a**) showed a burst release at 1 min and a full drug release after 30 min. The drug release profile of the Franz diffusion method (**b**) showed a sustained release of imeglimin from approximately up to 6 h and a complete drug release after 24 h. The results are presented as the mean ± SD (n = 3).

**Figure 6 pharmaceutics-15-01208-f006:**
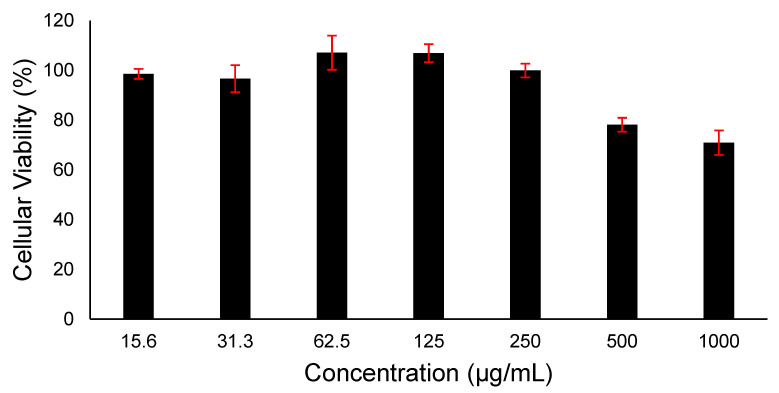
Cell viability of different doses of imeglimin upon 24 h HFF-1 cells exposure. The data demonstrated that imeglimin is safe at ≤250 µg/mL. Results are presented as mean ± SD (n = 3).

## Data Availability

The authors confirm that the data supporting the findings of this study are available within the article.

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
