# Peer review of "Development of Imeglimin Electrospun Nanofibers as a Potential Buccal Antidiabetic Therapeutic Approach"

_pharmaceutics, 2023, doi:10.3390/pharmaceutics15041208_

Round 1

Reviewer 1 Report

The manuscript titled: Development of Imeglimin Electrospun Nanofibers as a Potential Buccal Anti-Diabetic Therapeutic Approach is a study about the Development of electrospun nanofibers for buccal administration of anti-diabetic imeglimin. The study is interesting and well written, however some details need to be clarify before publication:

- I would like to know why authors choosed the pH of 6.8, in order to simulate the buccal microenvironment? Having in account that the pH of buccal mucosa is around 5.5 and 6.5.

- I recommend the authors to perform release studies in Franz diffusion cells and also permeation studies with porcine buccal mucosa.

- How the authors intend to retain the nanofibers in buccal mucosa without a mucoadhesive polymer in the formulation? Probably it will be a sublingual administration instead of buccal.

- I recommend the authors to perform citotoxicity studies with the drug loaded in nanofibers and nanofibers alone, in order to observe the toxicity of the system for buccal cells.

Author Response

The response is attached. Thank you!

Reviewer 2 Report

Alamer’s et al manuscript submitted to Pharmaceutics describes the preparation of the imeglimin-loaded PVP-electrospun nanofibers for buccal delivery. In general, the manuscript looks interesting and covers urgent topic. It is well organized and illustrated. The experimental part provided in quite details. However, there some issues I would like to address the authors to improve their manuscript and make it stronger.

Major:

1. The authors studied the cytotoxicity for 24 h using the HFF1-1 cells. First, the selection of this cell line for imeglimin testing is not well motivated so looks strange. Second, the longer assay (for 72 h) is required to exclude the delayed cytotoxicity. In addition, the IC50 values of this drug should be calculated for both 24 and 72 h.  

2.       Section 3.4. and Figure 4 describe the HPLC calibration for imeglimin detection. The authors should also show the example chromatogram(s) for this calibration curve. For example, as stack of several chromatograms with different loading of the drug to the column. Both chromatograms and calibration curve can be combined into Figure 4 as subfigures (a) and (b).

Minor:

1.       Legend to Figure 5 duplicate the text in the paragraph above the figure. It should be revised to exclude the result stating – just a heading. Instead, the details on release conditions (medium, component concentration, temperature, and so on), in which this curve was obtained, should be provided.

2.       It is better to use commonly accepted SI units for designation of time, namely, s, min and h, instead of seconds, minutes and hours. It should be corrected throughout the text.

3.       I would also recommend adding a structure of imeglimin somewhere to the text to better understand its properties (hydrophilicity/hydrophobicity, charge, etc).

Author Response

The response is attached. Thank you!

Reviewer 3 Report

Dear authors,

I read your paper entitled "Development of Imeglimin Electrospun Nanofibers as a Potential Buccal Anti-Diabetic Therapeutic Approach" and here are my comments:

1. Please add in the Introduction the novelty and added value of the paper with respect to existent literature.

2. I wouldn't say in the Introduction Imeglimin nanofibers as this is added within the formulation. The nanofibers are based on PVP.

3. SEM images: how can you distinguish blank and drug loaded nanofibers?

4. The number of references must be increased with some more relevant papers on the topic.

5. Physico-chemical analysis is not shown. FTIR or XPS couod be added to check some interactions between the drug and PVP nanofibers.

6. Discuss the physical (or chemical) interactions between the drug and PVP. Is the drug absorbed onto the surface? Do yoy have some drug entrapped within the nanofibers? This should be explained.

7. LIVE-DEAD test is important for biological aasay. The safety of the formulation is not suffciently asessed at in vitro scale.

8. What about in vivo testing?

9. Dissintegration test should be performed in simulated saliva. See some references here: Ali J, Bong Lee J, Gittings S, Iachelini A, Bennett J, Cram A, Garnett M, Roberts CJ, Gershkovich P. Development and optimisation of simulated salivary fluid for biorelevant oral cavity dissolution. Eur J Pharm Biopharm. 2021 Mar;160:125-133. doi: 10.1016/j.ejpb.2021.01.017. Epub 2021 Jan 30; Joseph Ali, Jong Bong Lee, Sally Gittings, Alessandro Iachelini, Joanne Bennett, Anne Cram, Martin Garnett, Clive J. Roberts, Pavel Gershkovich,
Development and optimisation of simulated salivary fluid for biorelevant oral cavity dissolution, European Journal of Pharmaceutics and Biopharmaceutics,
Volume 160, 2021, Pages 125-133 or similar.

10. Discuss the drug release mechanism related to the target application. Is this profile suitable?

11. Why did you choose PVP as the drug carrier? The MW is pretty high and polymer accumulation within the body was reported in many studies (starting with the 2nd Word War when used as plasma). It is recommended to use low MW for PVP in medical applications (<20 000 g/mole).

Author Response

The response is attached. Thank you!

Round 2

Reviewer 1 Report

Dear authors,

I recommend the publication of the manuscript in the present form, having in account that all the reviewers comments were adressed.

Best regards

Author Response

Thank you 

Reviewer 2 Report

In general, the authors have performed the required revision and provided explanations for some of the questions. However, I would suggest that the motivation for selecting HFF-1 cells be given not only in the answer to my question, but also in the text of the manuscript.

Author Response

Thank you for your constructive feedback

We added this statement and referenced the previous studies that used the mentioned cell line (lines 217-219):

'HFF-1 cells were exploited in this study since it was previously reported as an in vitro model for drug delivery research of the oral mucosal cavity [32,33,34].'